# Effects of Metformin in Heart Failure: From Pathophysiological Rationale to Clinical Evidence

**DOI:** 10.3390/biom11121834

**Published:** 2021-12-04

**Authors:** Teresa Salvatore, Raffaele Galiero, Alfredo Caturano, Erica Vetrano, Luca Rinaldi, Francesca Coviello, Anna Di Martino, Gaetana Albanese, Raffaele Marfella, Celestino Sardu, Ferdinando Carlo Sasso

**Affiliations:** 1Department of Precision Medicine, University of Campania Luigi Vanvitelli, Via De Crecchio 7, I-80138 Naples, Italy; teresa.salvatore@unicampania.it; 2Department of Advanced Medical and Surgical Sciences, University of Campania Luigi Vanvitelli, Piazza Luigi Miraglia 2, I-80138 Naples, Italy; raffaele.galiero@unicampania.it (R.G.); alfredo.caturano@unicampania.it (A.C.); erica.vetrano@unicampania.it (E.V.); luca.rinaldi@unicampania.it (L.R.); francesca.coviello@studenti.unicampania.it (F.C.); annadimarti@alice.it (A.D.M.); gaetanaalbanese@hotmail.it (G.A.); raffaele.marfella@unicampania.it (R.M.); celestino.sardu@unicampania.it (C.S.)

**Keywords:** type 2 diabetes, metformin, heart failure

## Abstract

Type 2 diabetes mellitus (T2DM) is a worldwide major health burden and heart failure (HF) is the most common cardiovascular (CV) complication in affected patients. Therefore, identifying the best pharmacological approach for glycemic control, which is also useful to prevent and ameliorate the prognosis of HF, represents a crucial issue. Currently, the choice is between the new drugs sodium/glucose co-transporter 2 inhibitors that have consistently shown in large CV outcome trials (CVOTs) to reduce the risk of HF-related outcomes in T2DM, and metformin, an old medicament that might end up relegated to the background while exerting interesting protective effects on multiple organs among which include heart failure. When compared with other antihyperglycemic medications, metformin has been demonstrated to be safe and to lower morbidity and mortality for HF, even if these results are difficult to interpret as they emerged mainly from observational studies. Meta-analyses of randomized controlled clinical trials have not produced positive results on the risk or clinical course of HF and sadly, large CV outcome trials are lacking. The point of force of metformin with respect to new diabetic drugs is the amount of data from experimental investigations that, for more than twenty years, still continues to provide mechanistic explanations of the several favorable actions in heart failure such as, the improvement of the myocardial energy metabolic status by modulation of glucose and lipid metabolism, the attenuation of oxidative stress and inflammation, and the inhibition of myocardial cell apoptosis, leading to reduced cardiac remodeling and preserved left ventricular function. In the hope that specific large-scale trials will be carried out to definitively establish the metformin benefit in terms of HF failure outcomes, we reviewed the literature in this field, summarizing the available evidence from experimental and clinical studies reporting on effects in heart metabolism, function, and structure, and the prominent pathophysiological mechanisms involved.

## 1. Brief History of Metformin and Its Interaction with Cardiovascular Outcomes

The medicinal properties of metformin (dimethyl biguanide), an antihyperglycemic drug introduced in Europe in 1957 and registered about thirty years later in the US, were already known in the Middle Ages when the French lilac *Galega officinalis* containing the active compound galegine (isoamylene guanidine), was used to treat people with intense urination [1,2]. Today, metformin is the most commonly prescribed medicament for the treatment of diabetes, taken by millions of patients worldwide daily, including those with HF [3]. Based on its glucose-lowering effectiveness, safety, favorable effect on body weight, and low cost, metformin is supported by many scientific associations as first-line therapy in patients with T2DM. Above all, indication for use was strengthened based on its assumed favorable and long-lasting benefits on CV morbidity and mortality, best demonstrated by the 10-year follow-up of the landmark UKPDS trial [4,5]. Over time, a huge number of publications have highlighted the CV benefits of metformin, and in 2006, a joint statement of the American Diabetes Association (ADA) and the European Association for the Study of Diabetes recommended this drug as the initial pharmacologic intervention in T2DM [6,7,8]. Still, nowadays, ADA maintains this position with the recommendation of continuing the drug until tolerated and not contraindicated [9].

Shortly after metformin approval in the US, HF was listed as an absolute contraindication for use since the “phantom” of lactic acidosis, a legacy of previous negative experience with phenformin and buformin, two biguanides abandoned since the 1970s [10]. In the following years, this dictate was largely ignored, also by virtue of the very low risk of lactic acidosis in clinical practice with an estimate of <10 cases per 100,000 patient-years [11,12]. At the same time, data were accumulated until conclusive evidence revealed that the benefits of metformin use in diabetic patients with HF outweighed the potential risk [13,14]. With this information, in 2006 the Food and Drug Administration (FDA) removed the absolute HF contraindication, although acute or unstable congestive HF remained in the label’s warning section.

The scenario has completely changed since the publication in recent years of a series of CVOTs providing a robust evidence-based CV beneficial effect of glucagon-like peptide-1 receptor agonists (GLP1-RAs) and sodium-glucose co-transporter-2 inhibitors (SGLT2-Is) in diabetic patients at increased risk [15,16]. Currently, even in the presence of high CV risk, ADA continues to support metformin as the first-line drug in T2DM, with consideration of concurrent therapy with SGLT2-Is in patients with HF or kidney disease and either SGLT2-Is or GLP1-RAs in patients with predominantly atherosclerotic CV disease (CVD) [9]. Instead, the European Society of Cardiology recommends SGLT2-Is or GLP1-RAs as first-line therapy replacing metformin in patients with target-organ damage or several CV risk factors, and in those with clinically manifest CV comorbid conditions [17]. This guideline discrepancy has raised an intense debate in the medical community.

The arguments for keeping metformin as a cornerstone drug for diabetes even in the presence of a high CV risk are more than one. In the first instance, long-term experience of use, reasonably acceptable side effects, and, more importantly, CV benefits have historically been proven for over twenty years. Ultimately, in all positive CVOTs conducted so far, the newer glucose-lowering agents operated in the background of therapy with metformin that might have influenced the CV outcomes.

The interaction between metformin with SGLT2-Is and GLP1-RAs is a not yet solved issue. A meta-analysis of six CVOTs with four SGLT2-Is by Neuen et al. indicated clear and consistent reductions in CV outcomes, irrespective of the baseline metformin therapy [18]. Similar results emerge from a systematic review and meta-analysis of trials with GLP-1 RAs [19]. Instead, in a supplemental analysis of the EMPA-REG OUTCOME trial, the relative risk reduction of CV death in metformin nonusers was 54% vs. 29% in metformin users, although with a P interaction of 0.07 [20]. Similarly, the relative risk reduction in the composite of CV death or HF hospitalization in the CANVAS Program was 36% in metformin nonusers vs. 12% in metformin users, with a nominally significant P interaction of 0.03 [21]. Because of these results, the suspicion arose that background metformin therapy may obscure the CV benefits of SGLT2-Is, an interpretation that should be done with extreme caution as it is right to do for results from post hoc subgroup analyses [22].

In this review, we aim to support the possible beneficial role of metformin in the management of diabetic patients with established HF and, mainly, its preventive relevance in patients at risk for developing HF, such as those with uncomplicated diabetes, prediabetes, or metabolic syndrome. For this purpose, we synthetized the available evidence from experimental and clinical studies reporting on metformin effects in heart metabolism, function, and structure, and the prominent pathophysiological mechanisms involved.

## 2. Heart Failure Epidemiology in Diabetes

Diabetes is a systemic disease characterized by both micro and macrovascular complications, as well as a wide range of extravascular complications, which weigh heavily on both patient health and national health systems [23,24,25,26]. HF has emerged as a very common CV complication of diabetes, with a 2.5-fold times higher incidence in diabetic than nondiabetic people, exceeding that of myocardial infarction (MI) or stroke, with a prevalence in diabetic patients aged ≥65 years as high as 22% [27,28,29,30]. As suggested by Lundbæk in 1954 who first introduced the concept of specific DM-related cardiomyopathy, this strict epidemiological relationship depends on the possibility of HF occurrence in T2DM patients even in the absence of other canonical HF risk factors, mainly atherosclerosis and heart ischemic injury. The descriptive terminology of “diabetic cardiomyopathy” is currently used to define myocardial dysfunctions in the presence of diabetes and in the absence of coronary artery disease (CAD), valve heart disease, and other conventional risks for CV diseases (e.g., hypertension, dyslipidemia, and alcoholism) [31]. If diabetes is a well-established risk factor for HF, strong evidence indicates HF as a risk factor for T2DM. Roughly 40 percent of hospitalized patients with HF enrolled in clinical trials suffered from diabetes and among those with HF with reduced ejection fraction (HFrEF) without a known history of diabetes, 25% had prediabetes, 13% unrecognized T2DM, and 10% developed new-onset diabetes during the next four years [32,33,34,35]. The relationship between HF with preserved ejection fraction (HFpEF) and diabetes is even closer as T2DM affects about 45% of affected patients and, when present, the health-related quality of life is worsened and the risk of hospitalization, CV mortality, and all-cause mortality increased [36,37,38]. Generally, when HF and diabetes coexist, the one condition synergistically and mutually worsens the prognosis of the other. Diabetes or insulin resistance in HF patients reduces the functional capacity, increases the risk of readmission for HF, and doubles the yearly mortality [39,40,41,42]. On the other hand, HF represents a very harmful complication of T2DM, since there is a frequent progression to end-stage requiring heart transplantation, and a 3-year mortality of 40%, 10-fold higher than that of non-HF diabetic patients [43,44].

These impressive epidemiological findings emphasize the crucial clinical value of preventing this CV complication.

## 3. Activation of AMPK by Metformin

The AMP-dependent kinase (AMPK) is a serine-threonine protein kinase ubiquitously expressed in nucleated mammalian cells with a crucial role in cellular energy homeostasis. Its malfunction or absence is associated with metabolic disorders, primarily insulin resistance, and could be involved in the pathogenesis of diabetes. It is structurally a heterotrimeric protein complex containing a catalytic α domain and two regulatory β and γ domains [45]. The increase in intracellular AMP/ATP ratio, such as during strong exercise, hypoxia, or nutritional deficiency, may activate AMPK through phosphorylation of the amino acid threonine (Thr172) on the catalytic α subunit [46]. This is the site predominantly involved in AMPK activation, but several other amino-acid residues can be phosphorylated and determine the same effect [47]. AMP binding to the regulatory γ subunit preserves AMPK from dephosphorylation by its three protein phosphatases, 2A, 2C, and Mg^2+^/Mn^2+^-dependent 1E [48].

Once activated by increased AMP cellular levels signaling a low-energy state, AMPK prompts a switch from ATP-consuming anabolic pathways to ATP-generating catabolic pathways in order to maintain energy homeostasis. The result is an inhibition of the synthesis of triglycerides and proteins, and stimulation of glucose transport, glycolysis, and fatty acid (FA) oxidation [49,50]. These effects arise from the phosphorylation of a multitude of downstream effectors involved in the modulation of a myriad of cellular processes. In addition to acute regulation of enzymatic activities, AMPK may adaptively reprogram the metabolism through transcriptional changes [50].

The mechanism of metformin action is not completely clarified in humans, but animal and cell culture experiments indicate an activation of AMPK mainly via inhibition of mitochondrial respiratory-chain complex I and increase in the phosphorylated- to unphosphorylated-AMPK ratio [51,52]. For this reason, the drug is classified among the indirect activators of AMPK also taking into account that metformin may block the AMP-deaminase [53,54]. In this case too, the result is an increase in the AMP:ATP ratio, which in turn induces AMPK activation. Ultimately, metformin mimics an imbalance between energy supply and use as it happens during fasting and exercise. It is noteworthy that metformin may also directly bind to AMPK subunits determining an increased assembly of the enzymatically active heterotrimeric complex and a greater accessibility for upstream kinases [55,56] (Figure 1).

## 4. Mechanisms of Beneficial Impact of Metformin on Heart Failure

The current knowledge supports a protective role of metformin against HF based on a series of complex and multidirectional pleiotropic actions. On the one hand, the biguanide may prevent systemic and coronary atherosclerosis by correcting multiple CV risk factors and reducing endothelial dysfunction, oxidative stress, and inflammation. In addition, it may mediate direct positive functional and structural effects preserving LV morphology and the systo-diastolic performance of the heart (Figure 2) [7,57].

### 4.1. Correction of CV Risk Factors

Metformin may positively influence the development and progression of atherosclerotic disease, and therefore of CAD-related HF, in people with or without T2DM, including subjects with type 1 diabetes [58]. Merely, the drug is able to correct some atherosclerotic risk factors. Blood pressure was lowered in salt-induced hypertension of spontaneously hypertensive rats and in non-diabetic patients, specifically those with impaired glucose tolerance or obesity [59,60]. As reported in HF patients who lost weight, metformin causes a decrease in food intake to 300 kcal/day and a loss in body weight that results in improvement of insulin sensitivity, attenuation of inflammation, and a series of beneficial changes in stroke volume, cardiac output, and myocardial oxygen consumption (MVO_2_) [61,62]. A decrease in blood levels of triglycerides and LDL cholesterol may contribute to atheromatosis prevention [63].

Regarding hyperglycemia, a beneficial impact of glycemic control on CV outcomes is reported during acute coronary syndrome [64,65,66]. Instead, clinical trials targeting more intensive glucose-lowering therapy obtained benefits only in long-term follow-ups such as UKPDS, but were not helpful if introduced late in the course of T2DM as demonstrated by three major trials (ACCORD, ADVANCE, and VA-DT) [67,68,69,70].

Likely, the main atheroprotection depends on insulin-sparing and sensitizing action of metformin that is able to correct the generalized insulin resistance inherent in obesity and T2DM, and the associated systemic dysmetabolic milieu with low-grade inflammation and oxidative stress. The resulting beneficial effects on the arterial wall, both on endothelial and smooth muscle cells, may protect vasculature from fibrosis and remodeling [71,72,73,74].

### 4.2. Direct Beneficial Actions of Metformin on Myocardium

The heart expresses the organic cation transporters OCT1 and OCT3, which are able to control the cellular uptake of metformin, and compelling experimental evidence documents direct effects of metformin on both isolated cardiomyocytes and the beating heart (Figure 1) [75].

Metformin has been proven to stimulate multiple cardioprotective pathways, mainly implemented by the activation of AMPK, whose subunits in the myocardium display two different isoforms, the α1 subunit mostly expressed in cardiac endothelial cells and the α2 subunits in cardiomyocytes [48,76]. At the heart level, AMPK may phosphorylate a plethora of metabolic enzymes, transporters of metabolites, and signaling molecules, involved in the regulation of energy metabolism, protein synthesis, mitochondrial health, autophagy, oxidative stress, inflammation, and so on [57,77,78]. As proofs, metformin-related improvements in myocardial structure and function are not seen in AMPK-deficient mice and as indicated by a limited metanalysis of five studies, the myocardiocyte levels of phosphorylated AMPK from metformin-treated animals were almost double compared to controls (Figure 3) [78,79].

#### 4.2.1. Effects on Myocardial Energy Metabolism and Efficiency

Although the cardiomyocytes of a healthy heart can flexibly use different substrates depending on availability and energy requirements, they prefer FA oxidation to fuel the high rate of ATP production needed for the continuous heart mechanical work, so much that 70–90% of necessary myocardial energy supplied by this source [80,81]. Instead, profound changes in cardiac energy metabolism take place during the usually long and complex evolution of HF, with a progressive decline in FA utilization and an initial up-regulation and subsequent reduction of glucose utilization [82]. The reduction in ATP production represents a constant feature of heart failure [83].

By activating AMPK, metformin induces a series of cardiomyocyte responses that ameliorate glucose utilization, mitochondrial respiration, and ATP synthesis resulting in a better systolic and diastolic effectiveness [79,84,85]. The main AMPK downstream mediators of these responses are the eNOS and the peroxisome proliferator-activated receptor-g coactivator-(PGC) 1α, a crucial regulator of cellular energy substrate metabolism [57,79].

It has been observed that the AMPK activation induced by metformin enhanced translocation of membrane GLUT4 and glucose uptake in insulin-resistant cardiomyocytes [86,87]. Moreover, in hypertrophied rat heart metformin intensified the glycolysis by activating the rate-limiting phosphofructokinase 1 (PFK1) [88].

Metformin treatment in vivo or added to the perfusate of ex vivo rat hearts subjected to a highly increased workload prevented the AMPK down-regulation and, preserving the glucose uptake–oxidation coupling, avoided the detrimental intracellular accumulation of glucose 6-phosphate. This latter has been found to be responsible for downstream activation of mammalian target of rapamycin (mTOR) and consequent endoplasmic reticulum stress (ER) and contractile dysfunction [89].

The insulin resistance peculiar to HF impairs the metabolic adaptability of the myocardium, leading to an unbalanced lipid and glucose uptake that favors lipid accumulation and lipotoxicity in cardiomyocytes [90]. The protective effects of metformin on cardiac lipid dysmetabolism were investigated in prediabetic rats affected by hereditary hypertriglyceridemia, a model characterized by ectopic lipid deposition including the heart. A reduction of myocardial stearoyl-CoA desaturase, a key lipogenic enzyme, and increased glucose and decreased FA oxidation in the heart were observed after metformin treatment [91]. Concomitantly, the accumulation of the lipotoxic intermediate diacylglycerols and lysophosphatidylcholine was attenuated. A human study observed a decreased cardiomyocyte accumulation of lipids associated with a myocardial reduction of insulin resistance markers (IRS-1 and IRS-2) and lipogenic factors (PPAR-α and SREBP-1c) in healthy hearts transplanted in T2DM recipients chronically treated with metformin [92].

Experimental evidence proves that a lower ATP synthesis with a reduced energy conversion in mechanical work directly impairs contractile function, thus representing the main culprits of heart failure [83]. In this landscape, mitochondria have a leading role as suggested by the degenerative changes in cardiomyocytes collected from HF patients, and the association between mitochondrial abnormalities and severity of LV dysfunction observed in a canine model of HF [93,94,95]. In addition, cardiomyocyte mitochondrial activity is closely coupled to MVO2 as oxidative phosphorylation meets 95% of cardiac cellular energy demand [93]. Hence, drugs specifically able to influence cardiomyocyte energetics operating in mitochondria as metformin, so-called mitotropes, provide an attractive tool to improve cardiac performance in HF [96].

Experimental and clinical studies show that metformin may successfully reverse mitochondrial abnormalities in cardiomyocytes, improving mitochondrial respiration and ATP synthesis, lessening mitochondrial respiratory uncoupling and myocardial MVO_2_, and increasing myocardial efficiency [84,97]. In a mice model of HF after myocardial infarction, metformin improved cardiac systolic function, reduced apoptosis of myocardial cells, and improved mitochondrial function. All effects were associated with up-regulation in myocardial tissue of deacetylase Sirtuin 3 (SIRT3) and activation of PGC-1α, two factors closely related to mitochondrial energy metabolism as already reported in previous studies [84,98,99].

In a recent randomized controlled trial (RCT) using PET and transthoracic echocardiography, a 3-month treatment with metformin added to optimal medical HF regimen in 36 non-diabetic patients with symptomatic EFrHF, determined a 17% reduction of MVO_2_ and a 20% increase in myocardial efficiency expressed as work metabolic index [97]. This energy-sparing effect was achieved in addition to β-blocker therapy, which may increase efficiency by 39% and reduce MVO_2_ by 24% [100]. Overall, by ameliorating glucose metabolism, mitochondrial function, and ATP production, and by attenuating lipotoxicity in cardiomyocytes, metformin is able to improve systolic and diastolic effectiveness.

#### 4.2.2. Myocardial Anti-Oxidative and Anti-Inflammatory Effects

Oxidative stress represents one of the most significant insults disrupting cardiomyocyte homeostasis in failing myocardium that can be effectively counteracted by metformin through an increase of NO synthesis, amelioration of mitochondrial function, and other mechanisms [101].

The heart expresses three isoforms of NO synthase, endothelial NOS (eNOS) expressed in endothelial cells but also in the cytoplasm of cardiomyocytes and neuronal (nNOS) and inducible NOS (iNOS), both mainly located in cardiomyocytes [85]. The nanomolar amounts of NO generated by eNOS and nNOS play a beneficial physiological role. Differently, iNOS produces NO in micromolar concentration, namely an excess of NO that impairing myocardial contractility may be deleterious for diastolic and systolic LV function [102,103]. In detail, excessively high cellular NO level activates, via the second messenger cGMP, both the protein kinase G that blocks the L-type calcium channels and consequently decreases the calcium influx into cells, and the cGMP-activated phosphodiesterase that breaks down the cAMP [103].

Metformin activates eNOS via the AMPK-dependent phosphorylation and association with heat shock protein-9 [104]. The increase in local generation of NO leads to inhibition of oxidative stress and apoptosis together with vasodilation, and thereby improves coronary blood flow, afterload, and LV function [105,106,107]. In addition, studies in vitro demonstrated that whereas IL-1 beta induces the cardiomyocytes mRNA expression of iNOS and its de novo synthesis leading to an excess of NO local production, metformin is able to suppress the lipopolysaccharide (LPS)-induced iNOS and NO production in monocytes [108,109].

Considering mitochondria as the primary energy-generating organelles but also, if dysfunctional, the main source of reactive oxygen species (ROS), the other barrier implemented by metformin against oxidative stress is the stimulation of mitochondrial biogenesis [110]. In a recent study on H9C2 cardiomyocytes, metformin exerted protective effects against high glucose-induced oxidative stress by activating AMPK and enhancing the expression of transcription factors related to mitochondrial biogenesis (PGC-1α, nuclear respiratory factor-1, and -2) and of mitochondrial genes such as NDUFA13, an indispensable assembly factor of complex 1, and manganese-dependent superoxide dismutase [111].

Previous observations report that exogenous ROS administration to H9c2 cardiomyoblasts stimulated the c-Jun N-terminal kinase (JNK), a well-known member of MAPKs [112]. In experiments in H9c2 cardiomyocytes, metformin administration inhibited the mitochondrial overproduction of ROS induced by high glucose plus hypoxia/re-oxygenation injury, via signaling mechanisms involving activation of AMPK and concomitant inhibition of JNK expression [113]. Concurrently, metformin mitigated the associated inflammatory response evidenced by the significant increase in mRNA of pro-inflammatory cytokines (TNF-*α*, IL-1*α*, and IL-6) [113].

Oxidative stress has been implicated in the activation of endoplasmic reticulum (ER) stress, a promoter of cellular autophagy and apoptosis in the diabetic heart [114]. The activation of AMPK using metformin may decrease the ER stress-induced cell death [115].

Closely linked to oxidative stress, inflammation is considered a driving mechanism in the pathophysiology of HF that may also be controlled by metformin, a drug with largely proven flogosis-inhibitory effects, independently of its blood glucose-lowering action and with therapeutic potential in clinical conditions other than T2DM [116,117,118,119]. Actually, the use of metformin is significantly associated with reduced circulating levels of inflammatory markers and cytokines in diabetic as well as in non-diabetic people with HF, indeed in COVID-19 diabetic patients [120].

Since a main mechanism linking oxidative stress to tissue injury is the ROS-stimulated inflammatory response, the anti-oxidative properties of metformin may per se attenuate inflammation. In addition, metformin seems to exert a direct anti-inflammatory action by inhibiting the nuclear factor ĸB (NF-ĸB) signaling through AMPK-dependent and independent pathways [121,122].

The modulation of the local inflammatory response triggered by MI is crucial to protect the infarcted cardiac tissue and to prevent HF development. Acute administration of metformin in isoproterenol-induced MI in rats may halt the inflammatory responses and inhibit the MI-associated LV dysfunction through the activation of AMPK and subsequent suppression of Toll-like receptor 4 (TLR4), a factor that activates the expression of several pro-inflammatory cytokine genes [123]. Similarly, an inhibitory effect on TLR4 signaling with reduced post-MI cardiac dysfunction was induced by chronic pretreatment with a low dose of metformin [124]. Of relevance, studies have demonstrated a pivotal role of TLR4 even in myocarditis and HF [125].

A recent study investigating the inflammatory response during MI demonstrated that metformin could activate autophagy and subsequently inhibit the NLRP3-mediated pro-inflammatory response of myocardial macrophages [126]. The same effects on the autophagy-NLRP3 axis mediated the anti-inflammatory and cardioprotective response to metformin observed in high glucose-treated cardiomyocytes and in diabetic mice [127]. These results are in line with previous experiments on diabetic OVE26 mice indicating that through the AMPK signaling pathway, metformin administered for 4 months could upregulate autophagy activity and prevent cardiomyopathy [128].

The role of a dysfunctional T-cell-mediated immune response in the onset and progression of CV disease is well documented and increasing evidence supports inhibition of T-cell-mediated inflammation by metformin [129]. Overall, in vitro and in vivo studies in animal models of chronic inflammation indicate that metformin exerts its anti-inflammatory effect through an AMPK-dependent modulation of the mTOR and the signal transducer and activator of transcription (STAT) 3 and 5 of T-cells [130].

Overall, several studies show the positive effects of metformin on oxidative stress and inflammation, two crucial processes strongly interconnected by a reciprocal worsening that threaten heart health both systemically and locally. In the last years, it has been reported that metformin may protect epicardial adipose tissue, a structure closely interconnected to the myocardium through a shared microcirculation that when enlarged takes on a harmful secretory profile of adipokines, which can damage the heart through inflammatory mechanisms [131,132]. It is believed that this benefit mainly involves HFpEF [133].

#### 4.2.3. Prevention of Remodeling

Metformin shows potent anti-remodeling properties linked to attenuation of myocardial hypertrophy and fibrosis that may preserve LV morphology and systo-diastolic performance, irrespective of glycemic status and the nature of myocardial insults. These beneficial actions have been well demonstrated in a pacing-induced HF model in dogs and in murine models of ischemia-induced cardiac injury or spontaneously hypertensive, insulin-resistant state. In these experimental models, AMPK activation by metformin repressed cardiac remodeling and rescued cardiac function [79,106,134].

Mammalian cardiomyocytes lose the ability to proliferate after birth; thus their growth is the only mechanism that may enlarge cardiac mass to increase the contractile function and reduce ventricular wall stress in response to a workload excess [135,136,137]. The development of cardiac hypertrophy is associated with metabolic reprogramming increasing protein synthesis and strongly involving AMPK [136,138]. In hearts from AMPK-α2 knockout mice, hypertrophy induced by phenylephrine is greatly accentuated, and human mutations in the AMPKγ subunit cause familial hypertrophic cardiomyopathy [139,140]. On the contrary, AMPK activation can inhibit some molecular mechanisms promoting cardiomyocyte cell growth such as mTOR [141]. In vitro studies confirm that pharmacological activation of AMPK by metformin can inhibit two pathways involved in heart regulation of protein synthesis, the eukaryotic elongation factor 2 (eEF2) and the p70 ribosomal S6 protein kinase (p70S6K) [142].

Metformin mitigated cardiac hypertrophy in mice subjected to chronic pressure overload, through an AMPK-mediated activation of the eNOS-NO downstream signaling pathway [143].

In AngII-induced hypertrophy of cultured cardiomyocytes, metformin exerted anti-hypertrophic effects associated with AMPK-induced blunting of AngII type 1 receptor (AT1R) up-regulation and of mitochondrial dysfunction through the SIRT1/eNOS/p53 pathway [144]. This effect is in accord with the results of a rodent study in which SIRT2 repressed the aging-related and AngII-induced cardiac hypertrophy, whereas the loss of SIRT2 expression blunted this protection against cardiac hypertrophy, an effect likely caused by the failure to activate the liver kinase B1-AMPK pathway [145]. Similarly, experimental models demonstrated that the activation of SIRT1 along with PGC-1α and AMPK, and the suppression of mTORC1 along with its upstream regulator Akt, exerted cardioprotective effects by promoting survival of cardiomyocytes over their growth [146].

Another study demonstrated that aldolase A was hyper-expressed in a mouse model of cardiac hypertrophy and that activating AMPK by metformin or AICAR this overexpression as well as the cardiomyocyte hypertrophy were prevented [147].

Other evidence supports those anti-hypertrophic actions of metformin not mediated by AMPK activation. Repressing mTOR signaling, metformin was able to protect mouse hearts against ventricular hypertrophy and dysfunction in both wild-type and AMPKα2 knockout mice exposed to transverse aortic constriction-induced hypertrophy [148]. Using primary cultured cardiomyocytes, a recent study demonstrated that metformin could halt the transcription of genes involved in hypertrophy response, such as endogenous atrial natriuretic factor and brain natriuretic peptide genes [149]. In a previous study, metformin suppressed the acetylation of histone H3K9 by inhibiting the histone acetyltransferase activity of p300 that has been involved in the development of HF [150].

In humans, a trial enrolling few patients (MET-REMODEL) documented a reduction in LV mass indexed to height in 68 dysglycemic not diabetic patients with CAD treated with metformin over a 12-month follow-up [151].

The other deleterious player of heart remodeling is the interstitial accumulation of collagen, an important promoter of increased stiffness, and impaired heart muscle relaxation.

Sasaki et al. demonstrated that metformin reduced myocardial fibrosis and improved cardiac performance through suppressing mRNA expression of the transforming growth factor (TGF)-β1 [134]. Another study in mice subjected to transverse aortic constriction demonstrated that metformin could modulate the expression of extracellular matrix protein genes. In this model, biguanide inhibited the TGF-β1 synthesis induced by pressure overload and the phosphorylation of Smad3 factor and its TGF-β1-stimulated translocation to the nucleus, leading to impaired collagen synthesis in cardiac fibroblasts [152]. Metformin determined an AMPK-mediated reduction in the expression of TGF-β1, basic fibroblast growth factor, and TNF-α in primary cardiomyocytes [153]. In accordance, a reduced post-MI scar formation occurred in AMPKα1 knockout mice suggesting the centrality of this kinase in cardiac fibroblast/myofibroblast biology [154].

In a study examining the angiotensin (Ang) II-induced differentiation of cardiac fibroblasts into myofibroblasts, a critical event in the progression of cardiac fibrosis and pathological cardiac remodeling, metformin decreased the Ang II-induced ROS generation in cardiac fibroblasts by inhibiting the activation of the protein kinase C-NADPH oxidase pathway [155].

As largely documented, metformin may directly impact myocardial hypertrophy and interstitial fibrosis, the two structural hallmarks in the physiopathology of HF. In addition, the reduced production of advanced glycation end products (AGE) by biguanide may contribute. AGEs are reported to induce crosslinks within and between the long-living protein collagen, thus increasing its resistance to enzymatic breakdown that leads to a permanent interstitial accumulation [28,156,157]. Interestingly, metformin may correct the significant increase of cardiomyocyte apoptosis and death characterizing the heart of diabetic patients and representing the primum movens for hypertrophic and fibrotic LV remodeling [31,158]. Furthermore, it is reported that the AMPK activation by biguanide prevented the apoptosis of cardiomyocytes during their incubation with H_2_O_2_ and reduced the number of dead cardiomyocytes in the hearts of dogs with experimental HF [134]. Other experiments confirmed this effect in the setting of chronic myocardial ischemia [159].

#### 4.2.4. Amelioration of Left Ventricle Function

Several preclinical studies document positive effects of metformin on cardiac performance, as in a genetic model of spontaneously hypertensive insulin-resistant rats, in rodent models of volume-overload HF with streptozotocin diabetes, in dogs with pacing-induced HF, and in the setting of post-ischemic HF after permanent left coronary artery occlusion [79,106,134,157,160,161,162,163]. In a rodent model of HFpEF, chronic treatment with metformin reduced the elevation in right ventricular pressure and the extent of pulmonary small artery hypertrophy [164]. In a brief meta-analysis of six studies in various experimental models of HF, metformin therapy was associated with an 8% higher LVEF than other antidiabetic drugs [78]. A mechanism possibly justifying these results may be the enhanced phosphorylation of the phosphokinase A sites in the N2B spring element of titin induced by metformin in a mouse model with an HFpEF-like phenotype [165]. This change may lower the cardiomyocyte stiffness, a typical finding of myocardial biopsies from patients with HFpEF, in association with cardiomyocyte hypertrophy and interstitial fibrosis [166].

On the contrary, the results of clinical studies are somewhat conflicting. In two studies in diabetic subjects, metformin predicted LV dysfunction. This result is questionable as treated patients were more obese than controls [167,168]. Instead, other investigations documented benefits on diastolic and systo-diastolic function, except in a study on advanced HFrEF [169,170,171]. Moreover, long-lasting metformin therapy was beneficial for improving diastolic function and delaying the progression of HFpEF in hypertensive diabetic people [172]. The results of the few and small placebo-controlled RCTs evaluating metformin in the setting of HF are rather disappointing. A 3–4 month therapy in HFrEF patients with prediabetes or insulin resistance does not improve resting LVEF and global longitudinal strain, nor exercise capacity [97,173]. In the GIPS-III trial, a prospective study evaluating the effect of 4 months of metformin treatment in 380 non-diabetic patients presenting STEMI and a mean LVEF of 54%, the LVEF and the diastolic function was not affected at the end of therapy, and not even 2 years later [174,175,176]. Evidence from the nine RCTs included in a recent systematic review only supports a metformin beneficial effect in improving MVO2 and reducing NT-proBNP levels in insulin-resistant or T2DM patients with HF, especially those without overt signs of CVD [177]. Instead, in a recent trial enrolling 54 non-diabetic patients with metabolic syndrome, metformin treatment on top of lifestyle counseling ameliorated diastolic dysfunction [178].

Overall, an amelioration of cardiac functional parameters by metformin may be more likely expected in patients with mild forms of HF, mainly HFpEF, no history of CV events, and after a long time chronic treatment.

## 5. Effects of Metformin on Adverse Consequences of HF in Clinical Studies

Overall, a huge number of studies on small or large populations reported a beneficial impact of metformin therapy on HF prognosis.

### 5.1. Effects on Total and HF Mortality

Several authors described the reduction of HF mortality compared to sulfonylureas, glucose-lowering regimens other than metformin, or lifestyle modification alone, as well as of all-cause mortality compared to sulfonylureas or not-metformin monotherapy [14,179,180,181,182,183,184,185,186,187]. Treatment with metformin determined an 18% lower risk of death in elderly patients discharged with diabetes and HF [13]. A significant reduction of mortality was obtained even in diabetic patients with incident HF who were metformin new users or already receiving oral antidiabetic agents, as well as in subjects with both newly diagnosed HF and diabetes, with respect to patients not exposed to any antidiabetic drug [14,177,182,188].

Meta-analyses of observational studies or RCTs confirm positive outcome results, such as improved survival with metformin monotherapy, lower CV mortality vs. sulfonylureas, and reduced all-cause mortality in metformin-treated diabetic patients [189,190,191]. In a recent meta-analysis including 40 studies comprising over a million diabetic patients, metformin reduced all-cause mortality as well as the incidence of CV events in people with HF and CHD better than sulfonylureas or non-medication [192].

In HFpEF patients, metformin reduced long-term mortality following admission for acute HF and all-cause mortality in subjects with poor glycemic control [193,194]. A recent systematic review and meta-regression analysis confirmed a significant reduction of mortality in HFpEF by metformin, even after adjustment for HF therapies including β-blockers and angiotensin II inhibitor ACE-Is [195].

Instead, in a cohort study involving diabetic patients with advanced HF (III and IV NYHA class), one-year survival in metformin-treated and non-metformin-treated patients was 91% and 76% respectively, a result not statistically significant after multivariate adjustment [171].

The lowering of mortality risk by metformin seems to be lower than ACE-I/ARB but greater than β-blockers and detectable even in diabetic patients usually excluded from clinical trials due to the presence of several comorbidities, such as renal insufficiency, MI, and obesity [182,184]. Accordingly, in a systematic review of observational studies, metformin treatment was associated with a mortality of 23% compared to 37% in controls (mostly on sulfonylurea therapy), across different classes of kidney dysfunction and EF [196,197].

Only **a** few meta-analyses documented a null effect of metformin on all-cause and CV mortality and on the risk of HF [107,108,109]. Others reported a troubling increased risk of death with the concomitant use of metformin and SU [189,190,191,192,198,199,200,201].

Interestingly, a significantly lower in-hospital mortality emerged in metformin-user diabetic patients with COVID-19 [202,203,204].

### 5.2. Risk of Admission for HF

Metformin reduced the risk of admission for HF compared to thiazolidinediones, sulfonylureas, or other non-metformin contemporary regimens, and once again versus sulfonylureas in T2DM patients who persisted in the use of biguanide despite reduced kidney function [13,14,181,198,205]. This benefit was confirmed by systematic reviews and meta-analyses [189,190,191,196].

In a retrospective observational study, the benefits of metformin in reducing HF exacerbations dissipated after a few days of non-use, and the cumulative drug exposure did not decrease the risk of HF-related exacerbation [206]. Otherwise, in another study metformin was associated with a lower risk of hospitalization for HF in a dose-response pattern with a beneficial effect especially significant when used for more than 2.5 years [207].

### 5.3. Risk of New-Onset HF

In some studies, chronic treatment with metformin linked with a lower risk of new-onset HF [208,209]. McAlister et al. estimated that 4.4 cases of HF per 100 treatment-years developed in patients using sulfonylurea monotherapy versus 3.3 cases per 100 years in those using metformin monotherapy [210]. Examining 232 meta-analyses evaluating ten classes of diabetes drugs, metformin appeared neutral with regard to all CV outcomes, but it might decrease the risk of major adverse CV events, comprising HF, compared with placebo or no treatment [211].

Although with the well-known limitations of observational studies and meta-analyses but considering the advantage of a huge amount of data, these studies collectively emphasize the benefits of metformin in HF outcomes and confirm its better effectiveness in **the** first stages of EF, particularly HFpEF, as emerged in preclinical studies.

## 6. Concluding Remarks and Future Perspectives

Today, a large number of people suffer from concomitant diabetes and HF and there will be even more in near future. It is therefore a relevant focus to identify the best pharmacological approach for glycemic control, also useful to both prevent and manage this CV complication in diabetic people.

Metformin is the current first choice for the great majority of patients with newly diagnosed T2DM requiring medical therapy. Based on the literature, the data suggest the ability of metformin to avert the development and/or progression of HF. This early inclusion in a therapeutic protocol of diabetes is a great opportunity to implement a preventive strategy against HF, such an advantage is not documented for other antihyperglycemic drugs, comprising the newer ones. For instance, the CVOTs randomized to SGLT2 inhibitors showing CV benefits included patients generally with advanced diabetes and a great burden of CVD. Only a recent observational study evaluating the SGLT2-I effectiveness compared to metformin for reducing CV events in T2DM patients, treatment-naïve in the preceding year, registered after a short-/mid-term follow-up of approximately 600 days an 18% reduction in CV events including IMA, HF, and stroke. However, the wide 95% CIs of many outcomes precluded a definitive conclusion on an eventual greater CV benefit for gliflozins [212].

As regards the management of established HF, choosing to treat diabetic patients with metformin is a sensitive matter given the robust data from CVOTs published since 2015 providing compelling benefits for new classes of antihyperglycemic agents. However, these studies had as comparator group an active medicament and not placebo. Moreover, the high prevalence of baseline metformin use in these trials may corroborate a role for biguanide as first-line therapy for T2DM with high CV risk.

Unfortunately, after over 60 years of clinical use and apart from the UKPDS, no large trial has specifically assessed the impact of metformin therapy on risk for HF development or on the prognosis of established HF among patients with T2DM. Nor do we have comparison data with SGLT2-Is apart from those that had as a primary outcome the change in HbA1c [213]. This gap is not proof that advantages do not exist for metformin. Moreover, we cannot ignore the deep experience with its use in clinical practice and the long-term evidence, lasting almost two decades, from a great harvest of data from experimental investigations providing mechanistic explanations that incessantly continue to emerge in the literature. Otherwise, the information on the long-term CV outcomes of SGLT2-Is and GLP1-RAs are lacking, and mechanistic studies are quite scarce since the objective difficulty for new medications is to equalize the extraordinary endurance for well over half a century of metformin.

To date, we know that, at a minimum, metformin is not harmful to diabetics with concurrent HF and that it may even be beneficial in reducing CV mortality and morbidity, in addition to the not negligible potential role in the prevention of aging-related diseases as well as cancer [214,215,216,217].

To dispel any doubt and make complete clarity on the metformin efficacy in HF diabetic patients, large long-term clinical trials powered to assess CV protection would be needed. This project is extremely difficult to implement for many reasons, primarily the lack of an economic profit, and unlikely will be conducted unless the initiative starts from governments.

Future research is awaited to address other not negligible issues. First, it is necessary to clarify definitively the CV interaction of metformin with SGLT-Is. Second, the metformin protection against HF is worthy of more extensive investigation even in people without concomitant diabetes, since its cardioprotective properties detached from antihyperglycemic effectiveness. In this matter, the ongoing DANHEART is the first study powered to address the effect of metformin (and of hydralazine-isosorbide dinitrate) in people with prediabetes or diabetes suffering from HFrEF [218]. Other data of interest will be provided by a large US trial evaluating the effects of extended-release metformin on clinical outcomes among nearly 8,000 patients with prediabetes and established atherosclerotic CVD, in which hospitalization for HF is a secondary outcome measure (VA-IMPACT; NCT02915198). This study was placed on temporary administrative hold since the COVID-19 pandemic (last update posted: 24 September 2021). Another placebo-controlled trial is evaluating the effects of prolonged-release metformin on the risk of CV events in patients with dysglycemia and high CV risk [219].

Finally, it can be appealing to explore if metformin may be an efficient therapeutic approach for the HFpEF phenotype, a condition notoriously linked to systemic metabolic or inflammatory diseases and for which no established therapies currently exist.

## Figures and Tables

**Figure 1 biomolecules-11-01834-f001:**
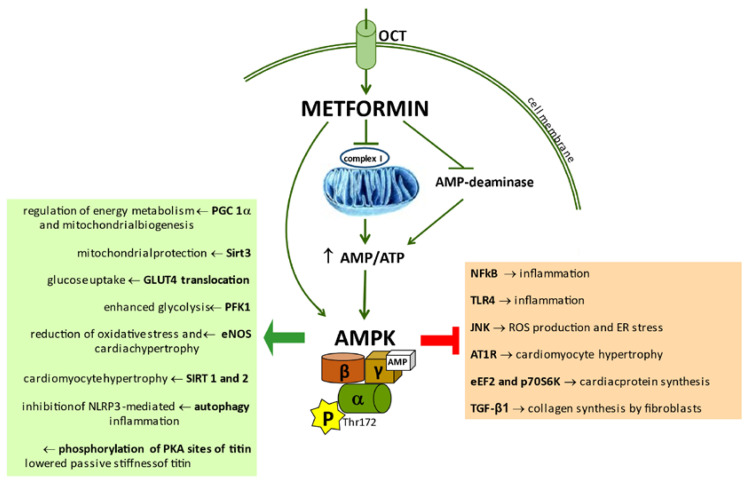
Activation of AMPK by metformin and its principal downstream pathways involved in HF physiopathology. AMPK: AMP-dependent kinase; AT1R: AngII type 1 receptor; eNOS: endothelial nitric oxide synthase; eEF2: eukariotic elongation factor-2 kinase; GLUT4: glucose transporter protein type-4; JNK: c-Jun N-terminal kinase; NF-ĸB: nuclear factor-kB; OCT: organic cation transporter; p70S6K: protein kinase 70S6; PFK1: phosphofructokinase 1; PGC1α: peroxisome proliferator-activated receptor-g coactivator-1α; PKA: protein kinase A; SIRT1-2-3: deacetylase Sirtuin 1-2-3; TGF- β: transforming growth factor- β; TLR4: Toll-like receptor 4.

**Figure 2 biomolecules-11-01834-f002:**
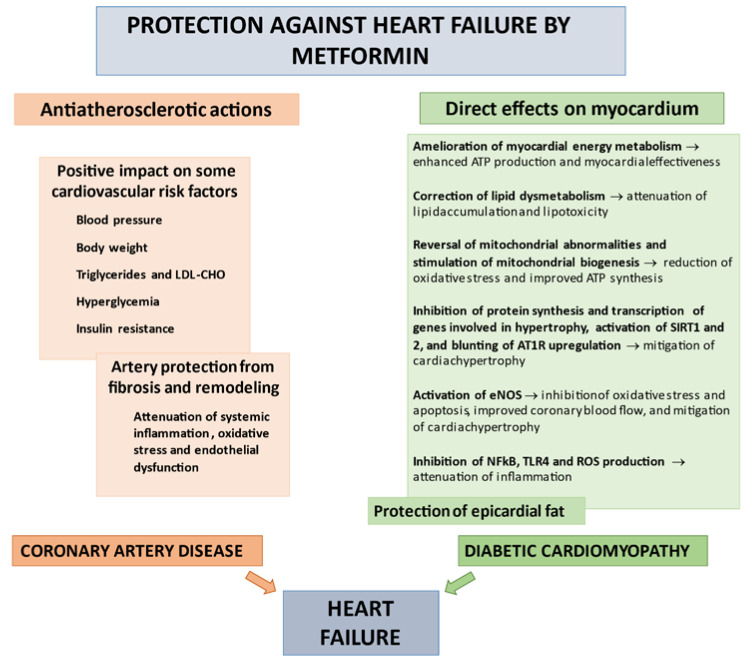
Anti-atherosclerotic actions and direct myocardial protective effects of metformin against HF. AT1R: AngII type 1 receptor; eNOS: endothelial nitric oxide synthase; LDL-CHO: low-density lipoprotein cholesterol; NF-ĸB: nuclear factor-kB; ROS: reactive oxygen species; SIRT1: deacetylase Sirtuin1; TLR4: Toll-like receptor 4.

**Figure 3 biomolecules-11-01834-f003:**
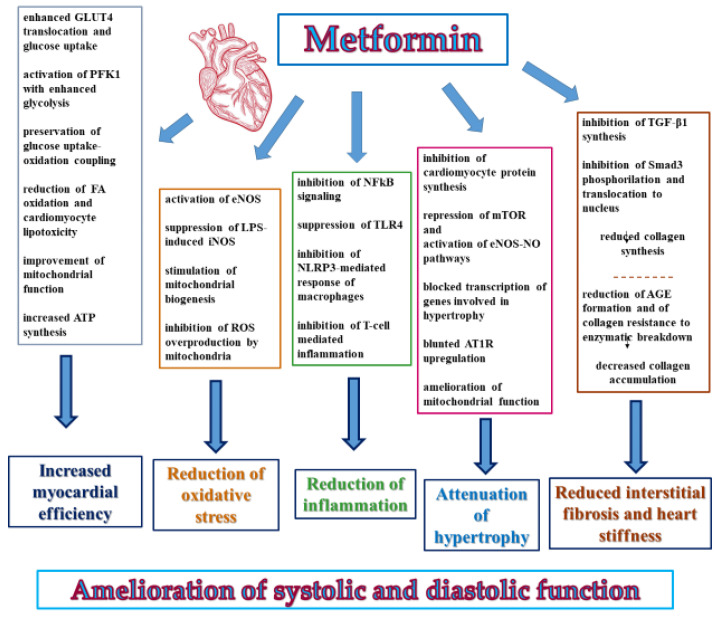
Main mechanisms of cardio-protection by metformin in failing heart (data from preclinical studies). AGE: advanced glycation endproducts; AT1R: AngII type 1 receptor; eNOS: endothelial nitric oxide synthase; FA: fatty acid; GLUT4: glucose transporter protein type-4; iNOS: inducible nitric oxide synthase; LPS: lipopolysaccharide; mTOR: mechanistic target of rapamycin; NF-ĸB: nuclear factor-kB; NO: nitric oxide; PFK1 phosphofructokinase 1; ROS: reactive oxygen species; TGF-β1: transforming growth factor β1; TLR4: Toll-like receptor 4.

## Data Availability

Not available.

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
