# Peer review of "Effects of Metformin in Heart Failure: From Pathophysiological Rationale to Clinical Evidence"

_biomolecules, 2021, doi:10.3390/biom11121834_

Round 1

Reviewer 1 Report

This review sought to clarify the puzzle of the relationship between metformin and heart failure. Although the authors have gathered hundreds of literature to demonstrate that metformin is beneficial to the heart by emphasizing the experimental and clinical studies evidence of cardiac metabolism, function, and even included the pathophysiological mechanisms. However, metformin is the first medicine that patients would take if has diabetes. It lowers down the A1C and delays prediabetes from becoming diabetes while diabetes is the top risk for heart failure. Hence, this review focuses on metformin's beneficial effect in preventing heart failure. It should cover or clarify several aspects:

  1. The current structure needs to be adjusted to clarify the central topic of this review.  several similar reviews on this topic have been published. What this review is trying to clarify to the readers should be more concise and be addressed. Metformin has a direct or indirect protective effect on heart failure. Whether metformin regulates the metabolism then indirectly affects the cardiac function or directly benefits the heart, needs to be included.
  2. At the end of each subsection, there should have a summary and guide the prospects for the future.
  3. the 'prospective' paragraph at the end of the paper should be provided. 

Author Response

The authors would like to thank the reviewer for his valuable comments which enabled us to improve the manuscript.

  1. The central topic has been better clarified in the aim of the review (see Introduction, lines 98-104), the structure of paper better arranged and the text significantly shortened (the little difference of total 571 lines vs 626 lines, depends on the addition of some sentences and paragraphs as required by reviewer 1). To this end, the section 4.1 and particularly the subsection 4.2.4 were substantially modified, with minor changes in the remaining text. A third figure (figure 2) has been included representing the direct effects of metformin on myocardium and the indirect ones by systemic actions of drug. Accordingly, the associated text has been modified (lines 166-171).

  1. A concise summary has been reported at the end of subsection 4.2.1 (lines 274-276), 4.2.2 (lines 347-353), 4.2.3 (lines 412-426), 4.2.4 (lines 457-459), and at the end of section 5 (lines 513-516).

  1. A paragraph entitled “Concluding remarks and future perspectives” (lines 519-571) has been provided.

Reviewer 2 Report

This review explains role of metformin in heart failure well with a lot of references. Metformin has many important functions for energy metabolism and regeneration of tissues. At first metformin was important as glucose lowering compound, but at present  it is gaining importance as stimulator of autophagy and ameliorating molecule for heart failure. From this point of view, this review is very important.

There are some questions and comments as follows.

  1. In line 139, AMP-dependent kinase should be changed to AMP-activated protein kinase.
  2. Both activation of SIRT1/PGC-1α and inhibition of mTORC1 shifts the balance of cellular priorities so as to promote cardiomyocyte survival over growth, leading to cardioprotective effects in experimental models (Packer, European Heart Journal, 2020, 41-3856-3861). Function of SIRT2 is mentioned in the text (line 426-427). If possible, SIRT1 function should be mentioned.

Author Response

The authors would like to thank the reviewer for his valuable comments which enabled us to improve the manuscript.

  1. “AMP-dependent kinase” changed to “AMP-activated kinase” (line 134).

  1. The function of SIRT1 has been added (lines 381-384) and the relative reference included (n.146).

Reviewer 3 Report

The manuscript entitled Effects of Metformin in heart failure: from pathophysiological rationale to clinical evidence”. The article’s topic sounds interesting but there are many mistakes in this manuscript that the authors need to revise. Please redraw the Figure 1 for a more detailed mechanism, because it is too brief.

Below are some comments for the authors to check.

Title page: Correspondence author uses the symbol *. Co-equal authors use the symbol +.

Page 1 Line 23: HF and T2DM, change to the full name when they first show.

Page 4, Figure 1: Authors must redraw the Figure 1. Because it is very important in this article for connecting the AMPK to downstream mechanism. OCT, add the full name in the figure legend. Change cellmembrane to cell membrane in the picture.

References section. The first word should be capitalized, and the other words should be lowercase, such Ref 3, 7, 9, 17, 22, 23, 34, 36, 37, 43, 48, 62, 66, 67, 69, 77, 82, 86, 87, 91, 92, 94, 96, 98, 102, 108, 109, 113, 126, 130, 134 to 138, 140, 145, 152, 155, 156, 163, 172, 173, 180, 186, 190, 193, 200, 204, 208, 210, 212, 221, 222 and 226. The authors need to revise the style of several references.

Author Response

The authors would like to thank the reviewer for his valuable comments which enabled us to improve the manuscript.

  1. Accordingly, we modified the text in the title page.

  1. The full name of HF and T2DM is reported in the Summary and when first showed.

  1. The figure 1 was enriched with the addition of principal AMPK downstream pathways involved in HF physiopathology. “cellmembrane” was changed to “cell membrane”, and the full name of OCT added in the legend figure.

  1. As requested, the reference section has been modified.

Round 2

Reviewer 1 Report

After the revision, this review reached to the satisfactory level.